# Adaptation of CD4 in gorillas and chimpanzees conveyed resistance to simian immunodeficiency viruses

Cody J Warren[†‡], Arturo Barbachano-Guerrero[†], Vanessa L Bauer, Alex Stabell, Obaiah Dirasantha, Qing Yang, Sara L Sawyer*

Department of Molecular, Cellular, and Developmental Biology, BioFrontiers Institute, University of Colorado, Boulder, United States

## eLife Assessment

This study presents an **important** finding on how lentiviral infection has driven the diversification of the HIV/SIV entry receptor CD4. Using a combination of molecular evolution approaches coupled with functional testing of extant and ancestral reconstructions of great ape CD4, the authors provide **solid** evidence to support the idea that endemic simian immunodeficiency virus infection in gorillas have selected for gorilla CD4 alleles that are more resistant to SIV infection. Expanding the study to interrogate the evolution and function of additional primate CD4 sequences could yield even stronger evidence.

**\*For correspondence:**
ssawyer@colorado.edu

[†]These authors contributed equally to this work

**Present address:** [‡]The Ohio State University, Columbus, United States

**Abstract** Simian immunodeficiency viruses (SIVs) comprise a large group of primate lentiviruses that endemically infect African monkeys. HIV-1 spilled over to humans from this viral reservoir, but the spillover did not occur directly from monkeys to humans. Instead, a key event was the introduction of SIVs into great apes, which then set the stage for infection of humans. Here, we investigate the role of the lentiviral entry receptor, CD4, in this key and fateful event in the history of SIV/HIV emergence. First, we reconstructed and tested ancient forms of CD4 at two important nodes in ape speciation, both prior to the infection of chimpanzees and gorillas with these viruses. These ancestral CD4s fully supported entry of diverse SIV isolates related to the viruses that made this initial jump to apes. In stark contrast, modern chimpanzee and gorilla CD4 orthologs are more resistant to these viruses. To investigate how this resistance in CD4 was gained, we acquired *CD4* gene sequences from 32 gorilla individuals of two species and identified alleles that encode 8 unique CD4 protein variants. Functional testing of these identified variant-specific differences in susceptibility to virus entry. By engineering single-point mutations from resistant gorilla CD4 variants into the permissive human CD4 receptor, we demonstrate that acquired substitutions in gorilla CD4 did convey resistance to virus entry. We provide a population genetic analysis to support the theory that selection is acting in favor of more and more resistant *CD4* alleles in ape species harboring SIV endemically (gorillas and chimpanzees), but not in other ape species that lack SIV infections (bonobos and orangutans). Taken together, our results show that SIV has placed intense selective pressure on ape *CD4*, acting to propagate SIV-resistant alleles in chimpanzee and gorilla populations.

## Introduction

Simian immunodeficiency viruses (SIVs) cause lifelong chronic infections in African monkeys and apes (*Klatt et al., 2012*; *Sharp and Hahn, 2011*). SIVs are classified in subfamily *Orthoretrovirinae* (genus *Lentivirus*) (*Coffin et al., 2021*). Currently, individuals from 45 different African primate species have

**Figure 1.** Overview of the emergence of simian immunodeficiency virus (SIV) into apes, ultimately giving rise to HIV-1. The figure shows, in the green box, the SIV reservoir that exists in African monkeys. Chimpanzees became infected with viruses from this reservoir resulting in the emergence of a new virus lineage, SIVcpz (*Bailes et al., 2003*; *Sharp et al., 2005*). From there, chimpanzees infected both gorillas and humans. The final two great ape species, orangutans and bonobos, are not known to harbor an SIV.

demonstrated antibodies that react to SIVs in serological tests, and in 37 of these species, SIV exposure has been confirmed by obtaining at least partial sequence of the actual virus (*Ayouba et al., 2014*). Complete genome sequence is available for only 27 SIVs (*Ayouba et al., 2014*). These viruses are named 'SIV' followed by a three-letter subscript that refers to the host primate species from which each was isolated (e.g., SIVcpz was isolated from chimpanzees) (*Ayouba et al., 2014*). Remarkably, of the tested primate species in Africa, approximately 90% of them have been associated with at least one SIV, indicating nearly pervasive infection of African primates (*Ayouba et al., 2014*).

HIV-1 emerged into humans from this diverse viral reservoir, but was not a spillover of virus directly from monkeys to humans. Instead, a key transition was the spillover of SIVs into great apes, which then set the stage for infection of humans (*Figure 1*). First, SIV of chimpanzees (SIVcpz) arose following the cross-species transmission and recombination of multiple SIVs from infected monkeys upon which chimpanzees predate (*Bailes et al., 2003*; *Sharp et al., 2005*). It is unknown if this virus recombination event occurred in the monkey reservoir before the first chimpanzee was infected or if it occurred within chimpanzee populations. Subsequently, SIVcpz transmitted to gorillas (giving rise to SIVgor) (*Van Heuverswyn et al., 2006*; *Takehisa et al., 2009*). Chimpanzees and gorillas have been endemically infected with SIVcpz and SIVgor since those spillover events (*Sharp and Hahn, 2011*). Spillover to humans from both chimpanzees and gorillas subsequently occurred on multiple occasions (*Van Heuverswyn et al., 2006*; *Keele et al., 2006*; *Plantier et al., 2009*). One of these spillovers yielded HIV-1 'group M' – the pandemic virus that has swept the globe, infecting over 80 million people. A third great ape species native to Africa, the bonobo, remains uninfected with SIV. Orangutans, the final great ape species, are native to Asia and also remain uninfected.

CD4 is the primary entry receptor for primate lentiviruses (SIV and HIV). CD4 is a surface protein expressed on T cells, where it is bound by the viral envelope (Env) glycoprotein to begin viral entry into the cell. To understand the role that CD4 plays in dictating the host tropism of SIVs, one must first appreciate the remarkable evolutionary signatures contained in the *CD4* gene. *CD4* has evolved under positive natural selection over the course of primate evolution (*Meyerson et al., 2014*; *Zhang et al., 2008*). This type of selection operates in favor of new alleles of *CD4* that have better resistance to virus entry (*Meyerson and Sawyer, 2011*). As such, it has been noted that most of the sequence evolution in *CD4* has been concentrated to the region encoding the D1 domain that interacts with HIV and SIV Env (*Meyerson et al., 2014*; *Zhang et al., 2008*). Even though natural selection operates at the level of alleles circulating within primate populations (*Ohainle and Malik, 2021*; *Russell et al., 2021*), the ultimate outcome is fixed *CD4* sequence divergence between species. As a result, we have demonstrated that different primate orthologs of CD4 vary dramatically in the lentiviruses that they will engage (*Warren et al., 2019a*).

Here, we focus on a key event in the emergence of HIV-1 into humans – the spillover of SIVs from monkeys to apes. First, we reconstructed and tested ancestral forms of CD4 at two important nodes in ape speciation, prior to the infection of chimpanzees and gorillas with these viruses. These ancestral CD4s fully support entry of diverse SIV isolates representing the viruses that made this initial jump to apes. In stark contrast, modern chimpanzee and gorilla CD4 orthologs are less supportive of entry by these viruses, consistent with natural selection having shaped *CD4* to resist infection in these species. Second, we investigated the subsequent spillover of SIV from chimpanzees to gorillas. We gathered *CD4* sequences from 32 gorilla individuals of two species, and identified alleles that encode 8 unique CD4 protein variants. We then identified variant-specific differences in susceptibility to SIVcpz entry (the virus that spilled over to gorillas). By engineering single-point mutations from resistant gorilla CD4 variants into a permissive human CD4 receptor, we demonstrate that these mutations are responsible for resistance to virus entry in gorillas. A population genetic analysis supports that selection is acting in favor of more and more resistant *CD4* alleles in gorillas. Taken together with similar analyses in chimpanzees (*Bibollet-Ruche et al., 2019*; *Warren et al., 2019b*), our results show that SIV has placed intense selective pressure on ape *CD4*, retaining and propagating SIV-resistant *CD4* alleles.

## Results

### Receptor-mediated resistance to SIV entry is a trait acquired during ape speciation

First, we wanted to know what ape CD4 was like before SIVs spilled over to apes and began to exert infection pressure on them. We used an alignment of *CD4* from diverse simian primates, and the program PAML (*Yang, 2007*), to infer ancestral *CD4* sequences at the base of the hominin and hominid clades, at the evolutionary positions shown with red and blue nodes in *Figure 2A*. The reconstructions of sequences at these ancestral nodes yielded *CD4* sequences that differ from human *CD4* by only two (hominin) and five (hominid) nonsynonymous substitutions. Only one of these changes mapped to the D1 domain of CD4 (N52S; *Figure 2B*). We then synthesized these extinct *CD4* genes. We transduced Cf2Th (canine) cells with retroviral vectors that stably integrated each of these *CD4* genes (hominin ancestral *CD4*, hominid ancestral *CD4*, human *CD4*, gorilla *CD4*, chimpanzee *CD4*, or an empty vector; *Figure 2—figure supplement 1*). All cell lines were also transduced to stably express human *CCR5*, a critical co-receptor for SIV and HIV entry.

We then tested these extinct and modern CD4 proteins for their ability to support viral entry mediated by SIVcpz Env. Since we do not know the actual genetic sequence of the first SIV(s) to infect chimpanzees, the best alternate strategy is to test a phylogenetically diverse set of extant SIVcpz strains (*Figure 2C*). We also tested HIV-1 strains that are embedded within the SIVcpz clade because these represent the virus spillovers from chimpanzees to humans. To generate pseudoviruses bearing SIVcpz and HIV-1 Env, different Env expression plasmids were co-transfected into 293T cells along with a plasmid encoding HIV-1ΔEnv-eGFP. The cell lines stably expressing various CD4 proteins and human CCR5 were then infected with each of these pseudoviruses. The percent of GFP+ (infected) cells was measured by flow cytometry and viral titers were calculated as transducing units per milliliter (TDU/mL). All tested pseudoviruses displayed similar levels of infection on cells expressing human or the ancestral CD4 proteins (*Figure 2D and E*). This suggests that ancestral versions of CD4 in apes were susceptible to primate lentivirus entry, just as human CD4 is known to be today. On the other hand, cells bearing chimpanzee and gorilla CD4s were generally less permissive to virus entry (green in *Figure 2D and E*). We conclude that CD4 was originally permissive to primate lentiviruses, but that selective pressures exerted by SIVs in the chimpanzee and gorilla lineages led to the retention of mutations that now confer resistance to primate lentivirus infection. This has not happened in humans where selective pressure by HIV-1 is relatively recent.

### *CD4* alleles circulating in gorilla populations differentially support entry of SIVcpz

Natural selection operates on individuals within populations, and only over time can the effects of this selection be seen in the divergence of gene orthologs between species. We and others have shown that multiple *CD4* alleles circulate in chimpanzee populations, which encode CD4 variants more resistant to SIVcpz entry than human CD4 (*Bibollet-Ruche et al., 2019*; *Warren et al., 2019b*).

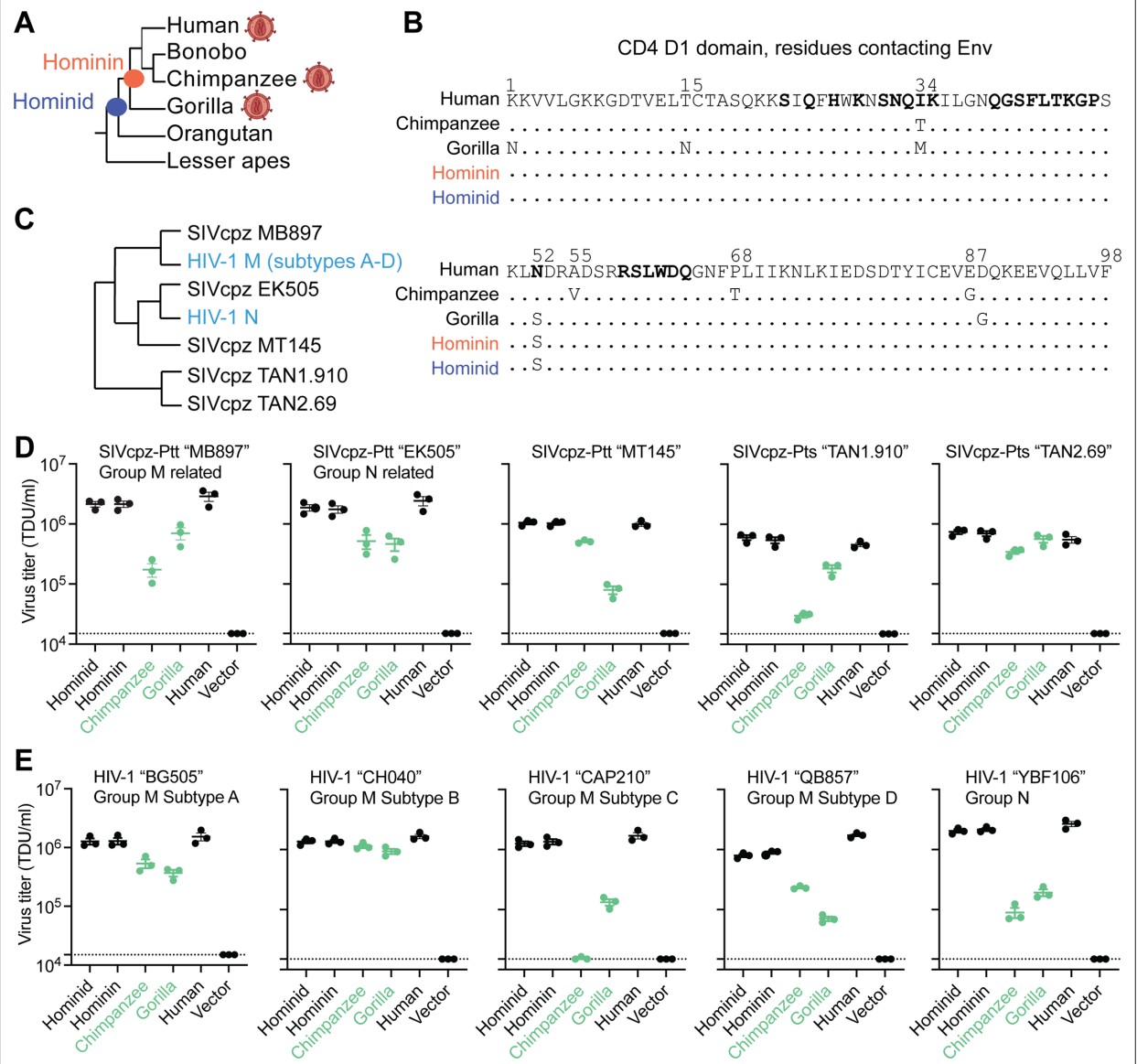

**Figure 2.** Receptor-mediated resistance to SIVcpz entry is a trait acquired in gorillas and chimpanzees. (**A**) Cladogram of ape species, highlighting the nodes for which ancestral *CD4* sequences were reconstructed. The virion diagram next to some ape species represents apes that are infected by simian immunodeficiency virus (SIV)/HIV. (**B**) An amino acid alignment of the CD4 D1 domain of human, chimpanzee, gorilla, and the inferred ancestral CD4 sequences. Dots represent identical residues compared to human and distinct amino acids and numerical positions are noted. Bolded residues on the human sequence represent sites known to directly interact with HIV-1 Envelope (*Liu et al., 2017*). (**C**) Cladogram of HIV-1 and SIVcpz based on previously published work (*Takehisa et al., 2007*), highlighting genetic relationships of the envelope (Env) clones used in this study. (**D, E**) HIV-1ΔEnv-GFP viruses were pseudotyped with Envs (top of graphs) from diverse (**D**) SIVcpz or (**E**) HIV-1 strains. Cf2Th cells stably expressing human CCR5 and various CD4s (X-axis) were infected with various volumes of these pseudoviruses and then analyzed by flow cytometry 48 h post-infection. GFP-positive cells were enumerated within the CD4/CCR5 positive gate and virus titers (transducing units per milliliter; TDU/mL) were determined for those samples falling within the linear infection range (n = 2 titration points). The mean virus titers obtained from each of three independent experiments were plotted (dots), with error bars representing the standard error of the mean (SEM). Dotted lines represent the lower limit of detection for this assay. SIVcpz-Ptt and SIVcpz-Pts refer to SIVs derived from the chimpanzee subspecies *Pan troglodytes troglodytes* or *Pan troglodytes schweinfurthii*, respectively.

The online version of this article includes the following source data and figure supplement(s) for figure 2:

**Source data 1.** Raw data associated with *Figure 2*.

**Figure supplement 1.** Flow cytometry gating strategy.

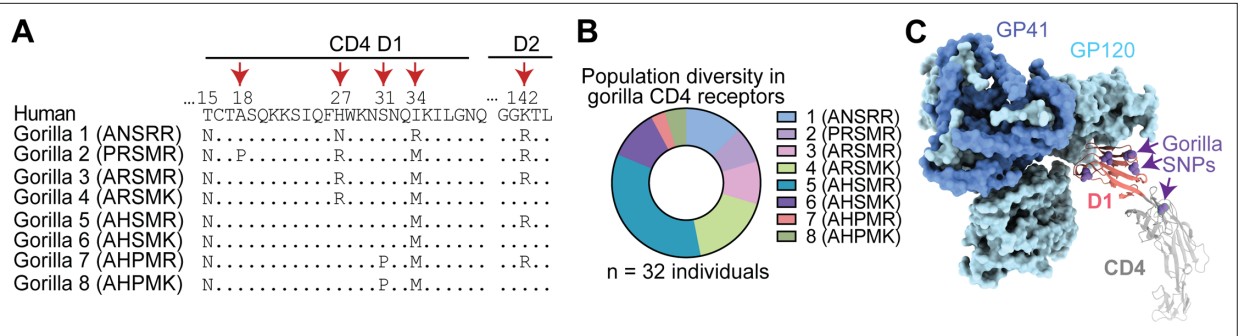

**Figure 3.** Identification of diverse gorilla *CD4* alleles. (**A**) Eight unique protein variants of gorilla CD4 were identified. The polymorphic sites (red arrows) are shown in the alignment, where dots indicate amino acid residues that are identical to human. (**B**) The frequencies of the eight alleles encoding unique CD4 protein variants are shown for three gorilla subspecies, *Gorilla gorilla gorilla* (n = 28), *Gorilla beringei graueri* (n = 3), and *Gorilla gorilla diehli* (n = 1). (**C**) Cryo-EM structure of an HIV-1 Env trimer in complex with human CD4 (PDB 5U1F) visualized in ChimeraX (*Goddard et al., 2018*). Individual gp120 and gp41 subunits are colored in light and dark blue, respectively. The CD4 D1 domain (red) and D2-D4 domains (gray) are shown, with gorilla nonsynonymous SNPs shown on the human sequence as purple spheres.

We next wanted to know if the same is true in gorilla populations. To similarly analyze gorilla *CD4*, we used the whole-genome sequence data from the Great Ape Genome Project (*Prado-Martinez et al., 2013*) to identify extant *CD4* alleles. We analyzed genetic data from 32 gorillas (*Gorilla gorilla gorilla* [n = 28]; *Gorilla gorilla diehli* [n = 1]; *Gorilla beringei graueri* [n = 3]) and found six nonsynonymous and five synonymous SNPs separating the individual alleles encoded. Five out of six of the nonsynonymous polymorphisms are located within the domain 1-encoding region of *CD4* (two are in the same codon, codon 27) and one in domain 2 (*Figure 3A*). A study of over 100 fecal samples from gorillas at field sites in Africa recently identified the same set of SNPs (*Russell et al., 2021*). In the definition of alleles considered here, we ignored synonymous polymorphisms. The nonsynonymous polymorphisms identified resulted in eight alleles that encode eight distinct CD4 protein variants. The frequencies of these eight alleles are heterogeneous, where allele 5 is the most common (*Figure 3B*). This, allele 5, was also the gorilla *CD4* that was tested in *Figure 2D and E* and shown in the alignment in *Figure 2B*. From looking at the sequences of these different alleles, we noticed a predicted glycosylation site (N-glycosylation tripeptide NXT) at position 15 that is fixed in the gorilla population but absent in the other African apes (*Figure 3A*). Interestingly, the gorilla *CD4* allele 2 codes for a proline at position 18, immediately after the tripeptide NCT, which strongly reduces the likelihood of glycosylation (*Gavel and von Heijne, 1990*). Since five of the six protein-altering polymorphisms are in the region corresponding to domain 1 of the CD4 protein, which directly binds to the lentiviral Env (*Figure 3C*), we next wanted to test their functional significance.

We made stable cell lines expressing each gorilla CD4 variant, along with human CCR5 (*Figure 2—figure supplement 1*). We then infected each of these with GFP pseudoviruses displaying envelopes from different strains of SIVcpz, as described above. We quantified the number of GFP+ cells to measure viral entry in cells bearing each CD4 variant. Again, we do not know the exact strain of SIVcpz that initially infected gorillas, so instead we have tested a phylogenetic diversity of SIVcpz strains. We found substantial differences in susceptibility to virus entry between the CD4 variants, varying by up to two orders of magnitude in some cases (*Figure 4*). All gorilla CD4 variants were equal to, or more resistant to infection than, the human CD4. We also tested pseudoviruses displaying a diverse set of envelopes from HIV-1 groups M and N, and found similar patterns (*Figure 4—figure supplement 1*). These data are consistent with SIV exerting selective pressure on gorillas in favor of resistant alleles of *CD4*. However, as would be expected in a host–virus arms race, the viruses are evolving too. As such, we found considerable differences on the entry phenotype for each SIVcpz strain evaluated, where a single host CD4 variant can be highly restrictive to one strain, while being fully functional for entry of another. As an outlier, SIVcpz TAN2.69 showed a uniformly strong ability to use any of the gorilla CD4 variants.

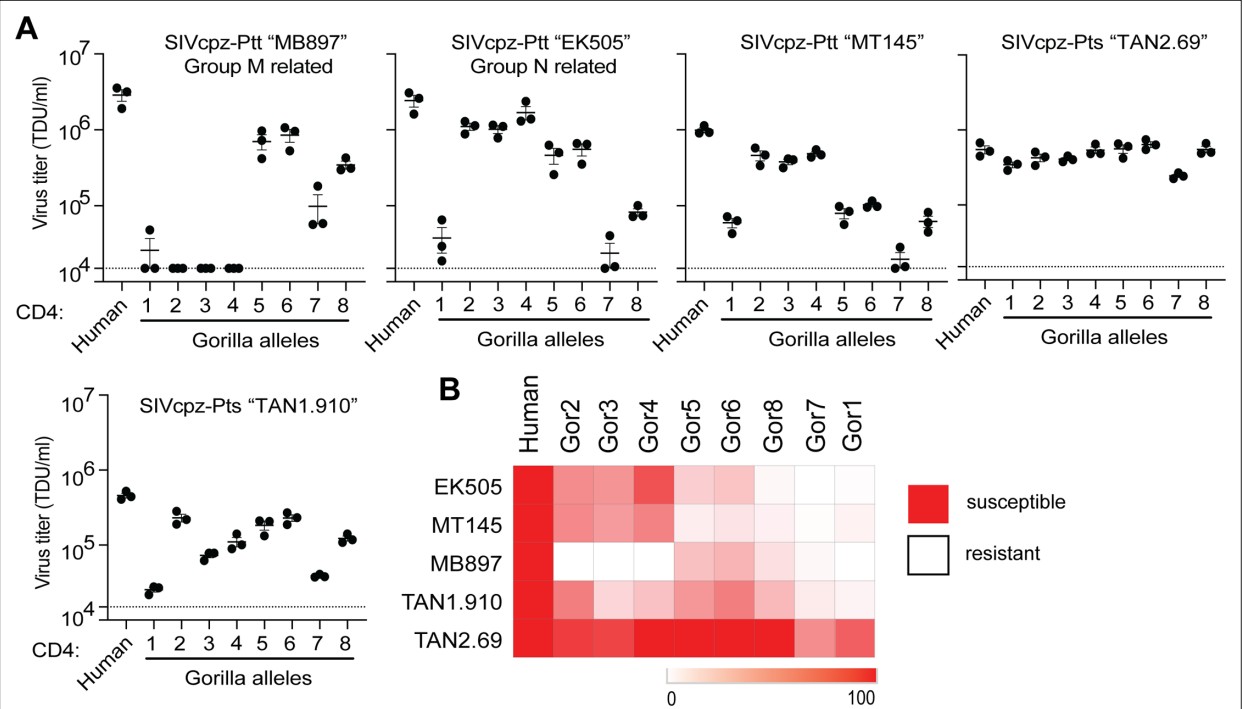

**Figure 4.** Gorilla CD4 variants differentially support entry of SIVcpz. (**A**) HIV-1ΔEnv-GFP viruses were pseudotyped with Envs (top of graphs) from diverse SIVcpz strains. Cf2Th cells stably expressing human CCR5 and various CD4s (X-axis) were infected with various volumes of these pseudoviruses and then analyzed by flow cytometry 48 h post infection. GFP-positive cells were enumerated and virus titers (transducing units per milliliter; TDU/mL) were determined for those samples falling within the linear infection range (n = 2 titration points). The mean virus titers obtained from each of three independent experiments were plotted (dots), with error bars representing the standard error of the mean (SEM). (**B**) Data from each pseudotyped Env in (**A**) were used to calculate virus titer means normalized to cells expressing human CD4 and were plotted as a heat map, where red and white represent susceptibility or resistance to viral entry, respectively.

The online version of this article includes the following source data and figure supplement(s) for figure 4:

**Figure supplement 1.** Gorilla CD4 alleles differentially support entry of HIV-1.

**Figure supplement 1—source data 1.** Raw data associated with *Figure 4*, *Figure 4—figure supplement 1*.

## Individual amino acid substitutions in gorilla CD4 protect against SIVcpz entry

We next sought to evaluate if CD4 polymorphisms found in gorilla individuals are protective when engineered into the human version of CD4, a widely susceptible receptor for primate lentiviruses. First, we investigated gorilla CD4 variant 2, which encodes a proline at position 18 that is predicted to prevent an otherwise fixed N-glycosylation at position 15 (*Figure 3A*). We noticed that gorilla variant 2 CD4 is highly susceptible to most of the SIVcpz strains tested in this study (*Figure 4*). Variant 3, which differs from variant 2 only by this proline, supported less entry by SIVcpz TAN1.910 due to this change in glycosylation status (*Figure 5A*). To explore the effects of this gorilla-specific glycan at residue 15, we generated cell lines stably expressing a mutated version of human *CD4* that encodes for the gorilla-specific glycosylation motif. We then challenged these cells with pseudoviruses displaying the envelope of different SIVcpz strains and consistently found a decrease in susceptibility to entry compared to wild-type human CD4 (*Figure 5B*). To confirm the glycosylation status of CD4, we performed CD4 western blotting on lysates from cells stably expressing each of the different versions of CD4. As expected, human T15N CD4, as well as gorilla variant 3, migrated at a higher molecular weight compared to human wild-type CD4 and gorilla variant 2, corresponding to the predicted number of glycans on CD4 domain 1 (*Figure 5C*). We then treated the lysates with PNGase F, an N-linked glycosidase, and found that all CD4 variants migrated to the same molecular weight, confirming that the mobility shift is due to the glycosylation status of CD4. Thus, most gorilla CD4 variants have gained a glycan at position 15 that reduces entry of SIV as compared to human CD4. This

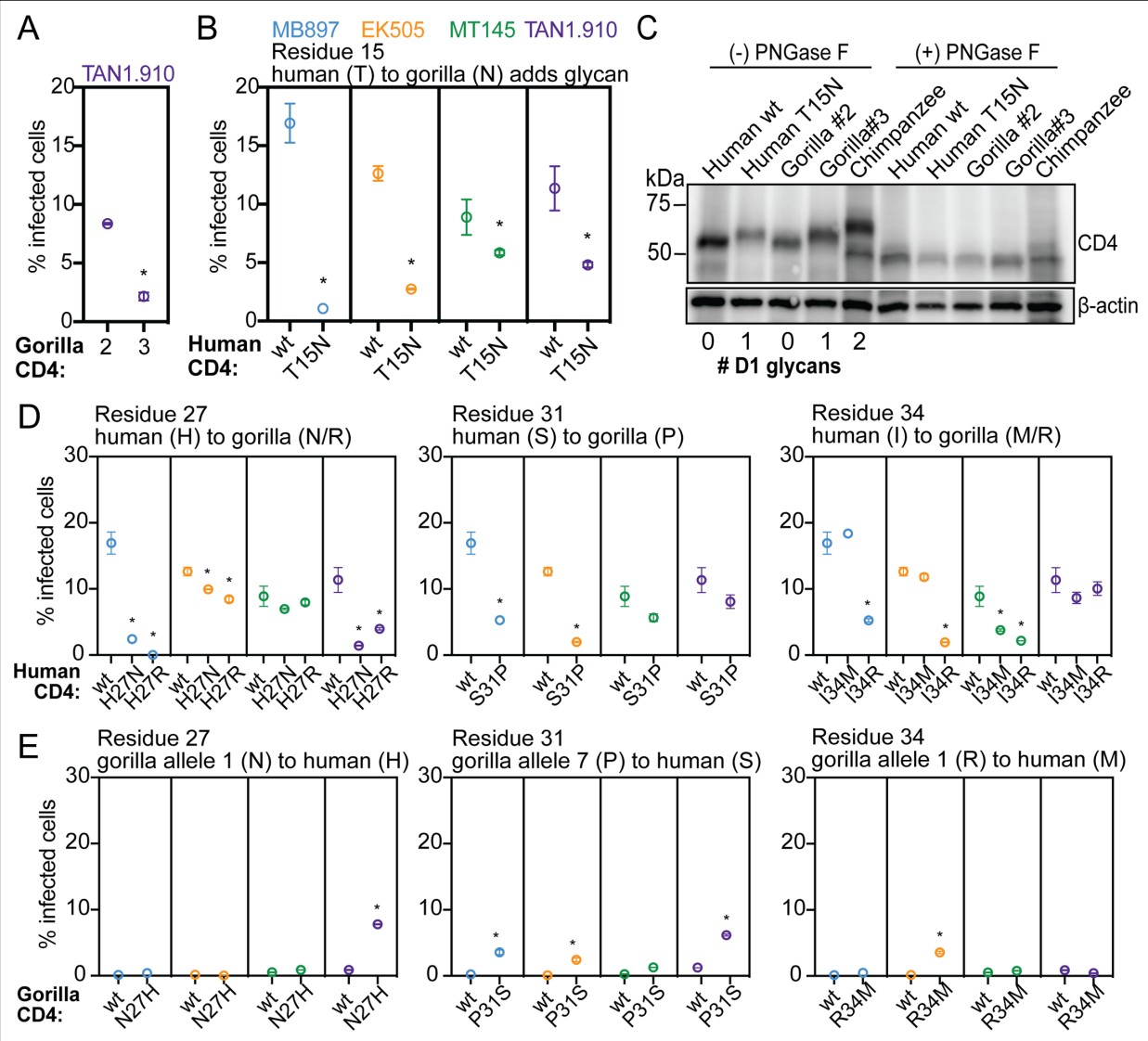

**Figure 5.** Individual amino acid substitutions that have occurred in gorilla CD4 protect against SIVcpz entry. (**A, B, D, E**) HIV-1ΔEnv-GFP viruses were pseudotyped with Envs from diverse SIVcpz isolates (MB897, blue; EK505, orange; MT145, green; TAN1.910, purple). Cf2Th cells stably expressing human CCR5 and wild-type (wt) or mutated human or gorilla CD4 (X-axis) were infected with these pseudoviruses and then the percent cells infected (GFP-positive cells) were enumerated by flow cytometry 48 h post infection. Data represent the mean ± SEM from two independent experiments, each with two technical replicates. Stars above data sets signify that both independent experiments showed significant statistical differences (p<0.05) when compared to wild-type by one-way ANOVA. (**C**) Lysates of Cf2Th cells stably expressing the indicated CD4 receptors in (**A**) and (**B**) were treated with PNGase F (to remove N-specific glycans) or left untreated and then probed for CD4 expression by western blotting. The number of N-specific glycosylation sites within the D1-domain of CD4 was determined computationally (*Gupta and Brunak, 2002*) and is shown under the blot. β-Actin served as a loading control. Images correspond to representative data from one of two independent experiments. Chimpanzee CD4, with two D1-domain glycans, has been described previously (*Warren et al., 2019b*).

The online version of this article includes the following source data for figure 5:

**Source data 1.** Raw data associated with *Figure 5*.

**Source data 2.** Western blot files uncropped label.

**Source data 3.** Western blot files uncropped raw.

phenotype is dependent on the lentiviral strain infecting. It seems that the gorilla *CD4* allele 2, which does not encode this glycan, would be at a fitness disadvantage. In support of this, allele 2 is one of the least frequent alleles in the gorilla population that we surveyed (*Figure 3B*).

We proceeded to test the amino acid residues at the other three polymorphic positions in domain 1 of gorilla CD4. We infected cells individually expressing mutant forms of human CD4 that coded for the gorilla-specific residues at positions 27, 31, and 34. We found that in all cases the mutant form of human CD4 encoding the gorilla-specific amino acid was significantly more restrictive to at least two of the four SIVcpz strains when compared to human wild-type CD4 (*Figure 5D*). In several cases, such as H27R CD4 expressing cells infected with SIVcpz MB897, we found drastic effects where a fully supportive receptor was rendered highly refractory to infection by a single amino acid substitution. We found that the protective role of these gorilla-specific substitutions was SIVcpz strain specific, demonstrating that collectively, the diversity found in gorilla individuals can confer relative protection to all the SIV strains we tested, but that SIVs are counter-evolving as well. These results suggest that single amino acid changes in domain 1 can drastically modify the interaction between CD4 and the lentivirus envelope, directly influencing virus entry.

To evaluate the reverse – if gorilla CD4 mutated to recapitulate the amino acids encoded in human CD4 may render the CD4 a better receptor for SIVcpz – we made and constructed cells expressing those CD4s and quantified the level of infection. We did not observe a full restoration of entry phenotype here and instead found only minimal increases in entry of SIVcpz through these receptors (*Figure 5E*). These results imply that the resistance to SIVcpz found in gorilla individuals is not dependent on single amino acids, but rather the cumulative effect of multiple amino acid changes. Overall, our data suggest that population-level sequence diversity in *CD4* of gorillas confers some level of protection against multiple SIVcpz strains.

## Positive natural selection has shaped *CD4* polymorphism in SIV-endemic ape species

Natural selection influences the frequency of alleles within populations. Alleles with deleterious effects will be kept at low frequency by purifying selection. On the other hand, alleles that confer a selective advantage will reach higher frequencies and/or be maintained in a population longer than expected due to different forms of positive selection (i.e., selective sweeps or frequency-dependent selection). We next tested how polymorphism in *CD4* has been shaped in ape species, with comparison made between apes that have been endemically infected with SIV (chimpanzees and gorillas) and those that have not (bonobos and orangutans).

Formal methods to detect the influence of selection on population-level nucleotide variation exist (*Fay and Wu, 2000*; *Fu and Li, 1993*; *Tajima, 1989*), but their statistical power is decreased in non-human ape species due to their small sample sizes and lower levels of variation (*Prado-Martinez et al., 2013*). Thus, we use a comparative approach to detect signatures of natural selection. To do this, we compared patterns of population-level diversity in *CD4* versus its neighboring genes. We compared SIV endemic apes (chimpanzee and gorillas) to apes uninfected (bonobos and orangutans) or recently infected on an evolutionary timescale (humans). This was performed for both nonsynonymous (protein altering) and synonymous (not protein altering) polymorphisms. Sequences for chimpanzee, gorilla, bonobo, and two orangutan species were obtained from the Great Ape Genome Project (*Prado-Martinez et al., 2013*) for *CD4* and 11 neighboring genes spanning 250 kb of the X-chromosome. For these same genes, human variation was obtained from the 1000 Genomes project. We calculated nucleotide diversity either based on the number of single-nucleotide polymorphisms (SNPs; Watterson's $\Theta_\omega$; *Watterson, 1975*) or mean pairwise difference between individuals ($\Theta_\pi$ ; *Tajima, 1983*). The mutation rate at *CD4* does not appear to be elevated given similar levels of variation between *CD4* and its neighboring loci when based on the number of SNPs ($\Theta_\omega$) (*Supplementary file 1* and *Figure 6—figure supplement 1*).

However, within the endemically infected species, nonsynonymous SNPs in *CD4* are at a significantly higher frequency compared to neighboring loci, represented by $\Theta\pi$ (*Figure 6A*; *Tajima, 1983*). This difference is not observed for synonymous variation. This discordance between nonsynonymous and synonymous variation suggests that the higher frequency of nonsynonymous variants at *CD4* in the endemically infected species is not explained by neutral or demographic evolutionary forces. In addition, the higher frequency of segregating nonsynonymous variation is restricted to the

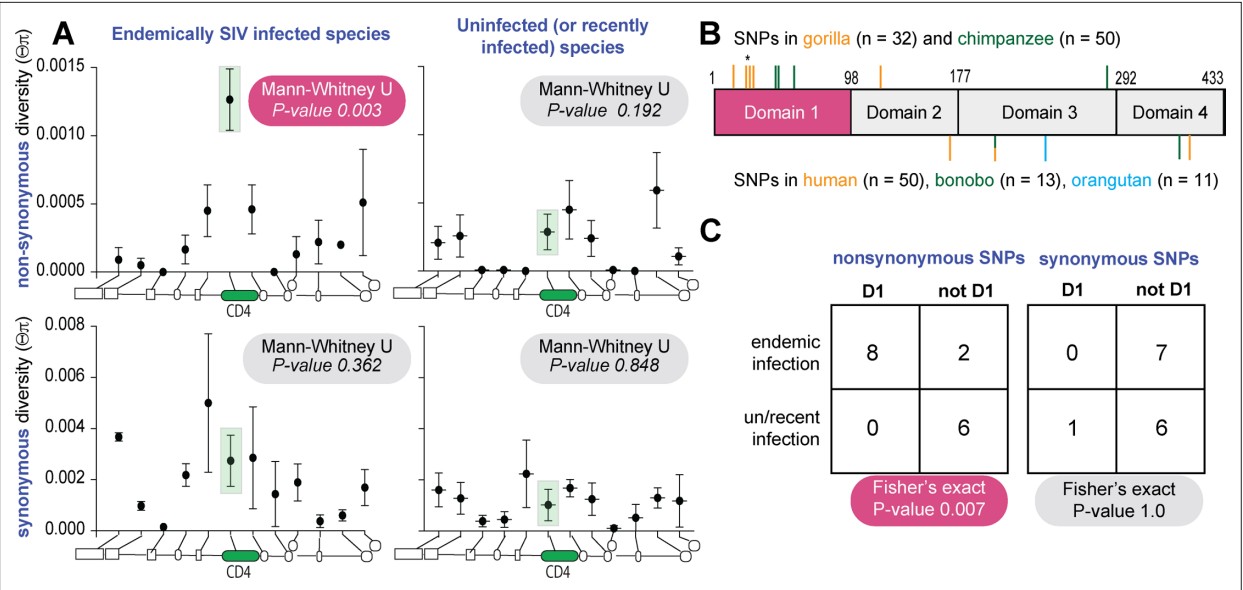

**Figure 6.** Positive natural selection has shaped *CD4* polymorphism in simian immunodeficiency virus (SIV)-endemic ape species. (**A**) Mean and standard error of mean of synonymous and nonsynonymous nucleotide heterozygosity (Θπ) at *CD4* and neighboring loci across species endemically infected with SIV (chimpanzee and gorilla) or un/recently infected (human, bonobo, and orangutans). Schematic along the bottom of each graph depicts the relative location of each locus as follows 5' to 3': *ZNF384, PIANP, COPS7A, MLF2, PTMS, CD4, GPR162, GNB3, CDCA3, TPI1, LRRC23*, and *ENO2*. Mann–Whitney test indicates whether heterozygosity at *CD4* is significantly different than neighboring loci. (**B**) Schematic of CD4 domain regions. Ticks above and below the CD4 box indicate the location of polymorphic sites for the infected and un/recently infected species groups, respectively. One of the polymorphic residues in gorilla contains two nonsynonymous changes in a single codon, marked by a star above the tick. (**C**) 2 × 2 contingency table and test results comparing synonymous and nonsynonymous polymorphism location relative to domain 1 between infected and un/recently infected species.

The online version of this article includes the following source data and figure supplement(s) for figure 6:

**Figure supplement 1.** Single-nucleotide polymorphisms (SNPs) in ape species.

**Figure supplement 1—source data 1.** Raw data associated with *Figure 6*, *Figure 6—figure supplement 1*.

**Figure supplement 2.** CD4 is under positive selection in primates.

endemically infected species. Taken together, these patterns are consistent with positive selection increasing the frequency of and/or maintaining nonsynonymous SNPs in *CD4* within the endemically infected species only. Also, in support of this, we find that gorilla and chimpanzee nonsynonymous polymorphic sites are significantly concentrated on the domain 1 of CD4 when compared to the un/recently infected species (*Figure 6B*). This difference is statistically significant (*Figure 6C*). These data suggest that long-term SIV infection in ape populations may be driving nonsynonymous SNPs to higher frequency in *CD4*, particularly in the region corresponding to domain 1 that directly interacts with the virus Env glycoprotein.

## Discussion

Pathogens are strong selective drivers of host gene evolution (*Demogines et al., 2012a*; *Meyerson and Sawyer, 2011*; *Warren and Sawyer, 2023*; *Warren and Sawyer, 2019c*). We and others have previously shown that the *CD4* gene has evolved under strong positive selection throughout the evolution and speciation of simian primates (*Meyerson et al., 2014*; *Zhang et al., 2008*). Selection on *CD4* is thought to be driven by the direct interaction between CD4 and the HIV/SIV envelope glycoprotein. Indeed, most of the sequence evolution in CD4 has occurred in the D1 domain that contacts Env (*Meyerson et al., 2014*; *Zhang et al., 2008*). We performed an updated analysis of positive selection in *CD4* including new *CD4* orthologs that have become available (*Figure 6—figure supplement 2*; *McBee et al., 2015*). We found that removing the D1 sequence from the analysis renders the gene no longer under positive selection. This sets the stage for the current study, which focused on selection on *CD4* within ape populations.

Within populations of animals, when alleles of *CD4* arise that can resist SIV, they would be predicted to rise in frequency. We and others have demonstrated that many *CD4* alleles circulating in chimpanzees convey an increased ability to restrict viral entry by SIVs (*Bibollet-Ruche et al., 2019*; *Warren et al., 2019b*). This is also observed for many other African primate species, where amino acid polymorphisms in CD4 resist viral entry (*Russell et al., 2021*). It is important to note that most African primate species currently harbor lentiviruses endemically, despite *CD4* evolution, meaning that CD4 remains functional for viral entry of at least some viruses despite the selective pressure to resist it. Therefore, we understand *CD4* to be evolving to convey natural tolerance in primates. 'Natural tolerance' refers to a species' ability to resist or tolerate a virus to an acceptable level for peaceful co-existence of the virus and host. It is obtained by evolutionary adaptations that occur over time, allowing the species to develop mechanisms to reduce the negative effects of the virus (*Pagán and García-Arenal, 2018*). For example, some species may evolve barriers (like resistant forms of CD4) that reduce the titers that a virus can achieve in their body. Studying natural tolerance is key to understanding virus reservoirs in nature.

Herein, we strengthen the insight into lentiviral tolerance via CD4 evolution in three ways. First, we reconstructed extinct ancestral forms of ape CD4 that pre-date SIV, and find that they were highly vulnerable to SIV entry. We then show that CD4 became less permissive to SIV in species that experienced long-term endemic infection. This resistant phenotype is associated with the accumulation of specific amino acid substitutions in the D1 domain of CD4. Second, we show that gorillas harbor a diversity of *CD4* alleles, all of which encode protein variants that are more resistant to SIV entry than is human CD4. Again, we demonstrate that these protein variants are gaining resistance by accumulating amino acid substitutions in the D1 domain, one of which creates a new motif for post-translational addition of a glycan to the CD4 protein. Protective (to the host) glycosylation of CD4 has recently also been observed by us and others in chimpanzees (*Bibollet-Ruche et al., 2019*; *Warren et al., 2019b*), and in another population sample of gorillas (*Russell et al., 2021*). Indeed, the evolutionary acquisition of a glycan shield on CD4 may be a recurring theme in the evolution of primate species that are plagued with SIVs (*Russell et al., 2021*). Lastly, using population genetics analyses, we show that nonsynonymous SNPs are enriched within ape species that are endemically infected with SIV (chimpanzees and gorillas) relative to those that are not (bonobos and orangutans) or which have been infected for less than 100 years (humans). This increased population-level diversity is observed only for *CD4*, and not shared by other genes neighboring the *CD4* loci.

Collectively, it is now clear that the sequence diversity (within species) and divergence (between species) of several primate genes have been driven by infection pressure from lentiviruses (*Daugherty and Malik, 2012*; *Demogines et al., 2010*; *Han et al., 2011*; *Johnson and Sawyer, 2009*; *Judd et al., 2021*; *Lee et al., 2012*; *Malfavon-Borja et al., 2013*; *Meyerson et al., 2018*; *Meyerson et al., 2015*; *Meyerson et al., 2014*; *Meyerson and Sawyer, 2011*; *Sawyer et al., 2007*; *Sawyer et al., 2006*; *Sawyer et al., 2005*; *Sawyer et al., 2004*; *Sawyer and Elde, 2012*; *Stabell et al., 2016*; *Warren et al., 2024*; *Warren and Sawyer, 2019c*; *White et al., 2014*; *Wilke and Sawyer, 2016*). The outcome of this virus-driven host evolution is ultimately detrimental to the very viruses that drove this evolution. When host genes like *CD4* become highly diverse within species, a given virus strain may only be able to infect a small number of individuals within the population. For instance, gorilla CD4 variant 1 is highly resistant to most of the SIVs we tested (*Figure 4*). In fact, we only found one SIV isolate, SIVcpz 'TAN2.69', which could enter cells through the receptor encoded by allele 1. This suggests that gorillas homozygous for allele 1 would largely be protected from most circulating SIV strains. Taking this example further, if allele 1 were to become fixed within gorilla populations, many strains of SIV in gorillas could go extinct.

In the long-term, the virus-driven evolution of genes like *CD4* also means that virus spillover between species – including the zoonotic spillovers that yield new human viruses – are less likely to happen. Indeed, a prevailing theme that has emerged in recent years is that receptor sequence divergence serves as a potent barrier to the movement of viruses between species (*Barbachano-Guerrero et al., 2023*; *Demogines et al., 2012b*; *Demogines et al., 2013*; *Hu et al., 2022*; *Kaelber et al., 2012*; *Kerr et al., 2015*; *Makin, 2022*; *Meyerson et al., 2014*; *Nahabedian et al., 2017*; *Ng et al., 2015*; *Sawyer and Elde, 2012*; *Warren et al., 2022*; *Warren and Sawyer, 2023*; *Warren and Sawyer, 2019c*). Specifically, CD4 and another host protein RanBP2 (*Meyerson et al., 2018*) played key roles in the transmission of SIVs to apes (see also *Sauter and Kirchhoff, 2019*). Further, this study

suggests that SIV entry is blocked by the CD4 receptor of some primate individuals that it might encounter. Therefore, spillover of lentiviruses between species will only happen when virus is transmitted between *key individuals* of two different species. The donor individual would need to have *CD4* alleles that yield high titers of SIV in its body, and the recipient individual would need to have *CD4* alleles that make it receptive to infection by this new virus.

## Materials and methods

### Ancestral reconstruction of the CD4 sequence at the base of the hominin and hominid clades

The ancestral state of CD4 was determined using the PAML software package as previously described (*Yang, 2007*; ; *Yang et al., 1995*). As input, we used an alignment of CD4 sequences from the following species: human (*Homo sapiens*; NM_000616.4), common chimpanzee (*Pan troglodytes*; NM_001009043.1), western lowland gorilla (*Gorilla gorilla gorilla*; XM_004052582.2), bonobo (*Pan paniscus*; XM_008973678.1), northern white-cheeked gibbon (*Nomascus leucogenys*; XM_004092147.1), Sumatran orangutan (*Pongo abelii*; XM_024256502.1), rhesus monkey (*Macaca mulatta*; NM_001042662.1), grivet (*Chlorocebus sabaeus*; XM_007967413.1), sooty mangabey (*Cercocebus atys*; NM_001319342.1), pig-tailed macaque (NM_001305921.1), crab-eating macaque (*Macaca nemestrena*; XM_005569956.2), gelada (*Theropithecus gelada*; XM_025401282.1), black snub-nosed monkey (*Rhinopithecus bieti*; XM_017891844.1), drill (*Mandrillus leucophaeus*; XM_011982990.1), Angolan colobus (*Colobus angolensis palliatus*; XM_011952091.1), golden snub-nosed monkey (*Rhinopithecus roxellana*; XM_010385914.1), and olive baboon (*Papio anubis*; XM_003905871.3).

### Genotype and allele determination of CD4 from gorillas

Short-read data available through the National Center for Biotechnology Information's (NCBI) Short Read Archive (BioProject PRJNA189439) were mapped onto the *G. gorilla* genome using BWA-MEM (*Li, 2013*). We applied GATK base quality score recalibration, indel realignment, duplicate removal, and SNP discovery and genotyping in each individual separately (*McKenna et al., 2010*). Joint genotyping and variant recalibration were performed in a species-specific manner and in accordance to the GATK best practices recommendations (*Van der Auwera et al., 2013*; *DePristo et al., 2011*). Variant recalibration was performed using SNPs called by the neighbor quality score method of ssahaSNP on capillary sequencing runs from NCBI's Trace Read Archive (*Ning et al., 2001*), dbSNP (if available), and high-quality SNPs called on the hg18 genome lifted over to the assembly used for mapping (*Prado-Martinez et al., 2013*). Processing was performed using custom scripts written in Python. Nucleotide sequence data reported are available in the Third Party Annotation Section of the DDBJ/ENA/GenBank databases under the accession numbers TPA: BK063765-BK063795.

### Receptor expression constructs and site-directed mutagenesis

Human (Genbank ID# MK170450) and chimpanzee (Genbank ID# NM_001009043.1) CD4 expression plasmids were constructed in a previous study (*Warren et al., 2019a*). The chimpanzee CD4 allele tested here is 'allele 6' as defined by us previously (*Warren et al., 2019b*), and has two glycans that impede virus binding to the receptor. Gorilla CD4 alleles and ancestral CD4s were commercially synthesized (IDT GeneBlocks) and gateway cloned into the pLPCX retroviral packaging vector (Clontech). Mutant versions of human and gorilla CD4 were constructed by standard site-directed mutagenesis methods using overlapping PCR primers encoding the modification. Both wild-type and mutant CD4 constructs were analyzed by Sanger sequencing prior to use.

### Generation of stable cell lines expressing CD4

HEK293T cells (ATCC CRL-11268) were cultured in DMEM (Invitrogen) with 10% FBS, 2 mM L-glutamine, and 1 X penicillin-streptomycin (complete medium) at 37°C and 5% $CO_2$. Cf2Th (ATCC CRL-1430) cells stably expressing human CCR5 (from *Warren et al., 2019a*) were cultured in complete medium supplemented with 250 µg/mL hygromycin. To produce retroviruses for transduction, HEK293T cells plated in antibiotic-free media ($1 \times 10^6$ cells per well in a six well plate) were transfected with 2 µg of pLPCX transfer vector containing the *CD4* gene of interest (or empty vector), 1 µg of pCS2-mGP (MLV gag/pol), and 0.2 µg of pC-VSV-G (VSV-G envelope) using a 3:1 ratio of TransIT-293 (Mirus) transfection

reagent to DNA according to the manufacturer's instructions. Forty-eight hours post-transfection, supernatant was collected, filtered through 0.22 μm cellulose acetate filters, and retrovirus stored at –80°C in single-use aliquots. Cf2Th cells stably expressing human CCR5 were plated at $2 \times 10^4$ cells per well of a 12-well dish (15% confluent) and 24 h later, transduced with 500 μL of retroviral supernatant by spinoculation at $1200 \times g$ for 75 min in the presence of 5 μg/mL polybrene. Forty-eight hours post-transduction, the cells were placed in complete medium containing selection antibiotics (250 μg/mL hygromycin and 3 μg/mL puromycin) and cultured until stable outgrowth was noted (>1 week). Stable cell lines were maintained indefinitely in selection media. To confirm expression of CD4, cells were analyzed by flow cytometry (*Figure 2—figure supplement 1*). Briefly, cells were harvested from culture plates, washed two times with PBS, fixed in 2% paraformaldehyde, and washed two times in flow buffer (1X PBS, 2% FBS, 1 mM EDTA). Fixed cells were stained for 30 min at 4°C with PerCP-Cy5.5 mouse anti-human CD195 (CCR5, BD Biosciences 560635) and AlexaFluor647 mouse anti-human CD4 (BD Biosciences, 566681), and analyzed using a BD Accuri C6 Plus flow cytometer (BD Biosciences).

## HIV/SIV envelope clones used in this study

Envelope clones for HIV-1 and SIVcpz EK505 and MB897 were constructed in a previous study (*Warren et al., 2019b*). SIVcpz MT145, TAN1.910, and TAN2.69 molecular clones were a gift from Brandon Keele (Frederick National Laboratory for Cancer Research, Frederick, MD) and used as template for PCR amplification. The RevEnv cassettes of SIVcpz were amplified by PCR using the following primer pairs, where the lowercase sequence corresponds to an added Kozak sequence for enhanced translation: MT145 (JN835462) forward 5'-tcgccaccATGGCAGGAAGAAGCGAGGG AGACG-3', reverse 5'-TTAAAGCAAAGCTCTTTCTAAGCCTTGT-3'; TAN1.910 (AF447763.1) forward 5'-tcgccaccATGGCAGGAAGAGAAGAGGACGC-3', reverse 5'- TTAATTTAAGGCTAGTTCCAGACC C-3'; TAN2.69 (DQ374657.1) forward 5'-tcgccaccATGGCAGGAAGAGAAGAGGACGC-3', reverse 5'-TTAATTTAAGGCTATTTCTAGACCCTGT-3'. PCR products were cloned into the pCR8/GW/TOPO TA plasmid (Thermo Fisher) and then shuttled into a Gateway-converted pCDNA3.1 mammalian expression vector (Invitrogen).

## Single-cycle HIV and SIV pseudovirus infections

To produce HIV-1ΔEnv-eGFP reporter viruses, $13 \times 10^6$ HEK293T cells were seeded into a 15 cm dish in antibiotic-free media and 24 h later transfected with 13.25 μg of Q23ΔEnv-GFP (group M backbone; *Humes and Overbaugh, 2011*) and 6.75 μg of envelope plasmid. Forty-eight hours post-transfection, the cell supernatant was harvested, concentrated (~100-fold) using Amicon Ultracel 100K filters (Millipore), and stored at –80°C in single-use aliquots. Cf2Th cells stably expressing CD4 and CCR5 were plated at $3 \times 10^4$ cells/well of a 48-well plate 24 h before infection. The cells (~80% confluent) were then infected with HIV-1 pseudoviruses in three different volumes (*Figures 2 and 4*), or a volume corresponding to 10–20% infection of cells expressing human CD4 (*Figure 5*). Infections were carried out by spinoculation at $1200 \times g$ for 75 min in the presence of 5 μg/mL of polybrene. Forty-eight hours post-infection, the cells were harvested from the plate and fixed in 2% paraformaldehyde. Fixed cells were washed three times with PBS and resuspended in 50 μL flow buffer (PBS buffer containing 2% FBS and 1 mM EDTA) and stained for 30 min at 4°C with the following antibody mixture: PerCP-Cy5.5 mouse anti-human CD195 (CCR5, BD Biosciences 560635, RRID:AB_1937312) and AlexaFluor647 mouse anti-human CD4 (BD Biosciences, 566681, RRID:AB_2744416) and analyzed using a BD Accuri C6 Plus flow cytometer (BD Biosciences). Following singlet cell discrimination, gates were drawn to capture double-positive cells expressing CD4 and CCR5, and then the percent of GFP+ cells was enumerated within that population. The data from ~$2 \times 10^4$ cells per technical replicate were analyzed using FlowJo v10. To calculate virus titers (*Figures 2 and 4*), the linear range of the infectivity curve was determined, and two points within the linear range were selected to calculate the mean virus titer in TDU/mL. The limit of detection for the titer calculation corresponds to a value of 0.2% GFP-positive cells. TDU/mL mean values were normalized to the titer of infection in cells expressing human CD4, and data used to construct a heat map using the Morpheus server (https://software.broadinstitute.org/morpheus); rows and columns were hierarchically clustered by Euclidean distance.

Statistical comparisons were performed between percentages of infected cells in some cases. Values of technical replicates of each biological replicate were compared between mutant and

wild-type CD4 versions by one-way ANOVA. If a statistically significant difference was found (p<0.05) in both independent biological replicates, an asterisk was added to the mutant column in the dot plot.

## Glycosylation state of CD4 by western blotting

Cf2Th cells stably expressing CD4 cells were lysed in Nonidet P-40 buffer (150 mM NaCl, 50 mM Tris·HCl pH 7.4, 1% Nonidet P-40 substitute, 1 mM DTT, 1 µL/mL Benzonase [Sigma-Aldrich #E1014], and protease inhibitor mixture [Sigma-Aldrich, #11873580001]) by resuspending the cell pellet and rocking at 4°C for 30 min. Cell lysate was cleared by centrifugation at maximum speed for 15 min. Whole-cell extracts were quantified using the BCA assay and 10 µg was subjected to PNGase F (New England Biolabs, #P0705S) treatment according to the manufacturer's protocol, including a paired sample with no glycosidase as control. Treated whole-cell extracts (5 µg per lane) were resolved on a 12% TGX Stain-free polyacrylamide gel (Bio-Rad, #1610185) by applying 180 V until loading dye ran off the gel. Protein was transferred to a PVDF membrane (MilliporeSigma, #IPVH07850) using a wet transfer apparatus set at 100 V for 60 min. The membrane was incubated with blocking buffer (tris-buffered saline 1X, Tween-20 0.1%, 5% nonfat dried milk) for 60 min at room temperature. Primary antibodies were diluted in blocking buffer and incubated with the membrane overnight at 4°C (1:1000 anti-CD4, Abcam #ab133616, RRID:AB_2750883). After primary antibody incubation, the membrane was washed 4 × 5 min in TBST (0.1% Tween-20). Secondary antibodies were diluted in blocking buffer and incubated with the membrane for 60 min at room temperature (1:10,000 anti-rabbit-HRP, Promega #W401B). After secondary antibody incubation, the membrane was washed 4 × 5 min in TBST (0.1% Tween-20), developed using ECL reagent (Sigma-Aldrich, #GERPN2232), and imaged on a Bio-Rad ChemiDoc Imaging System. As a loading control, membranes were reblotted to detect β-actin expression using primary (Cell Signal #3700) and secondary-HRP (Promega #W402B) antibodies and developed as described.

## Analysis of population-level selection acting on CD4

To compare the pattern of molecular evolution at CD4 relative to neighboring loci, we pulled population-level re-sequencing data for loci located within 100 kb downstream and upstream of CD4. Primate sequences were obtained from the Great Ape Genome project (*Prado-Martinez et al., 2013*) and size-matched (n = 50, matched with the number of chimpanzee sequences) human sequences were randomly selected to represent diverse ethnic groups from the Human 1000 Genomes Project, selecting 10 individuals for each of the five superpopulations (Africans [AFR], Admixed Americans [AMR], East Asians [EAS], Europeans [EUR], and South Asians [SAS]).

To identify the individual-specific SNPs within the selected loci, genotype data in variant call format (VCF) was directly downloaded from the International Genome Sample Resources (https://www.internationalgenome.org/) and the Great Ape Genome Project (https://eichlerlab.gs.washington.edu/greatape/). For human variants, the variant calls were made based on human reference genome annotation hg38, and individual-specific haplotypes were extracted by altering the reference sequence with the alternative SNPs annotated in the VCF files via a Perl script (https://github.com/santiagosnchez/vcf2fasta; *Sanchez-Ramirez, 2024*). For non-human primate variants, the short read genome sequences were mapped to human reference genome hg19 to generate the VCF files containing species- and population-level variants, as previously described (*Prado-Martinez et al., 2013*). The SNPs in the VCF files were further filtered by the variant call quality (GQ ≥15). Like the human sequences, the individual-specific haplotype sequences are re-constructed by correcting the reference sequence with VCF annotations.

In total, we obtained population-level variation for CD4 plus 15 other loci (six upstream and nine downstream). Four loci were removed from analysis because they have previously been shown to directly interact with a viral protein (*USP5* and *SPSB2*; *Jia et al., 2020*; *Rathore et al., 2020*; *Wang et al., 2019*; *Zhang et al., 2021*) or non-human primate sequencing reads did not map well with the human reference due to repetitive sequence (*LAG3* and *P3H3*). Coding loci included in this study (in order 5' to 3') are *ZNF384*, *PIANP*, *COPS7A*, *MLF2*, *PTMS*, *CD4*, *GPR162*, *GNB3*, *CDCA3*, *TPI1*, *LRRC23*, and *ENO2*. This was done for great ape species endemically infected with immunodeficiency viruses (chimpanzee and gorilla) and those newly or not infected (human, bonobo, and Sumatran and Bornean orangutans).

Sequences were aligned for each species individually using the Muscle alignment program (*Edgar, 2004*). DnaSP (*Rozas et al., 2017*) was used to haplotype-phase the downloaded sequences and to calculate levels of nucleotide diversity for each locus. Rarely we would observe an internal stop codon within a locus' reading frame. In these cases, both haplotypes for that individual were removed from analysis. We analyzed the subspecies of gorilla and chimpanzee together. While there is evidence of genetic differentiation between these subspecies (*Prado-Martinez et al., 2013*), this should not affect our comparisons as the differentiation is expected to be similar across all loci.

## Analysis of positive selection of CD4 in primates

### Sequence alignments

*CD4* sequences were aligned to the longest human isoform in MEGA X for macOS (*Stecher et al., 2020*) using the ClustalW alignment tool. Multiple sequence alignments were visually inspected, duplicate gene sequences were removed, and the gene isoform from each species that best aligned to the human reference was retained for further analysis. The terminal stop codon was removed, and aligned DNA and protein sequences were exported as fasta files. Codon alignments were generated using PAL2NAL (*Suyama et al., 2006*). Species cladograms for use in PAML were constructed following the species-level phylogenetic relatedness of primates (*Perelman et al., 2011*). Cladograms were generated using Newick formatted files and viewed with Njplot version 2.3.

## Evolutionary analysis

Codon alignments and unrooted species cladograms were used as input files for analysis of positive selection using the PAML4.8 software package (*Yang, 2007*). To detect selection, multiple sequence alignments were fit to the NSites models M7 (neutral model, codon values of dN/dS fit to a beta distribution bounded between 0 and 1), M8a (neutral model, similar to M7 but with an extra codon class fixed at dN/dS = 1), and M8 (positive selection model, similar to M8a but with the extra codon class allowed to have a dN/dS >1). A likelihood ratio test was performed to assess whether the model of positive selection (M8) yielded a significantly better fit to the data compared to null models (model comparisons M7 vs. M8 and M8a vs. M8). Posterior probabilities (Bayes Empirical Bayes analysis) were assigned to individual codons with dN/dS values >1. To calculate the posterior mean of $\omega$ over a sliding window, the per-site $\omega$ value was extracted from the M8 model, and the average $\omega$ value within the designated window size (80 amino acids) was calculated across the open reading frame in a sliding manner. With the window slide 1 amino acid each time to calculate the smoothed mean $\omega$ values.

## Materials availability

Materials, where available, can be requested directly to the corresponding author.

## Acknowledgements

This work was funded by the NIH (DP1-AI-175471, DP1-DA-046108, R01-AI-137011, R01-OD-034046 to SLS; T32 A1007447-25, F32 GM125442, and K99 AI151256 to CJW). The flow cytometry work was performed at the BioFrontiers Institute Flow Cytometry Core supported by NIH Grant S10ODO21601.

## Additional information

### Competing interests

Qing Yang, Sara L Sawyer: is a co-founder of, and holds financial interests in, Darwin Biosciences, Boulder, CO. The other authors declare that no competing interests exist.

### Funding

| Funder | Grant reference number | Author |
| --- | --- | --- |
| National Institutes of Health | DP1-AI-175471 | Sara L Sawyer |

| Funder | Grant reference number | Author |
| --- | --- | --- |
| National Institutes of Health | DP1-DA-046108 | Sara L Sawyer |
| National Institutes of Health | R01-AI-137011 | Sara L Sawyer |
| National Institutes of Health | R01-OD-034046 | Sara L Sawyer |
| National Institutes of Health | T32 A1007447-25 | Cody J Warren |
| National Institutes of Health | F32 GM125442 | Cody J Warren |
| National Institutes of Health | K99 AI151256 | Cody J Warren |
| National Institutes of Health | S10ODO21601 | Sara L Sawyer |

The funders had no role in study design, data collection and interpretation, or the decision to submit the work for publication.

### Author contributions

Cody J Warren, Conceptualization, Data curation, Formal analysis, Supervision, Investigation, Methodology, Writing – original draft, Funding acquisition, Writing – review and editing; Arturo Barbachano-Guerrero, Conceptualization, Data curation, Formal analysis, Investigation, Visualization, Methodology, Writing – review and editing; Vanessa L Bauer, Conceptualization, Formal analysis, Investigation, Visualization, Methodology; Alex Stabell, Conceptualization, Data curation; Obaiah Dirasantha, Formal analysis, Investigation, Visualization, Writing – review and editing; Qing Yang, Conceptualization, Formal analysis, Investigation, Visualization, Writing – review and editing; Sara L Sawyer, Conceptualization, Supervision, Funding acquisition, Project administration, Writing – review and editing

### Author ORCIDs

Cody J Warren ⓘ https://orcid.org/0000-0003-4101-2705
Arturo Barbachano-Guerrero ⓘ https://orcid.org/0000-0001-7483-839X
Vanessa L Bauer ⓘ https://orcid.org/0000-0003-0225-3215
Sara L Sawyer ⓘ https://orcid.org/0000-0002-6965-1085

Reviewer #1 (Public Review): https://doi.org/10.7554/eLife.93316.3.sa1
Reviewer #2 (Public Review): https://doi.org/10.7554/eLife.93316.3.sa2
Author response https://doi.org/10.7554/eLife.93316.3.sa3

## Additional files

### Supplementary files

Supplementary file 1. Levels of nonsynonymous and synonymous variation at CD4 and 11 neighboring loci. For each locus, we list the sample size and number of segregating nonsynonymous and synonymous SNPs for each species included in this study. We also include two measurements of nucleotide variability. One based on the number of SNPs ($\Theta_\omega$) and the other based on the frequency of each SNP ($\Theta_\pi$).

MDAR checklist

### Data availability

All data generated or analysed during this study are included in the manuscript and supporting files; source data files have been provided for *Figures 2, 4–6*.

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
