## [Editor Report · eLife Assessment]

This study presents an **important** finding on how lentiviral infection has driven the diversification of the HIV/SIV entry receptor CD4. Using a combination of molecular evolution approaches coupled with functional testing of extant and ancestral reconstructions of great ape CD4, the authors provide **solid** evidence to support the idea that endemic simian immunodeficiency virus infection in gorillas have selected for gorilla CD4 alleles that are more resistant to SIV infection. Expanding the study to interrogate the evolution and function of additional primate CD4 sequences could yield even stronger evidence.

---

## [Referee Report · Reviewer #1 (Public Review)]

Summary:

Authors previously demonstrated that species-specific variation in primate CD4 impacts its ability to serve as a functional receptor for diverse SIVs. Here, Warren and Barbachano-Guerrero et al. perform population genetics analyses and functional characterization of great ape CD4 with a particular focus on gorillas, which are natural hosts of SIVgor. They first used ancestral reconstruction to derive the ancestral hominin and hominid CD4. Using pseudotyped viruses representing a panel of envelopes from SIVcpz and HIV strains, they find that these ancestral reconstructions of CD4 are more similar to human CD4 in terms of being a broadly susceptible entry receptor (in the context of mediating entry into Cf2Th cells stably expressing human CCR5). In contrast, extant gorilla and chimpanzee CD4 are functional entry receptors for a narrower range of HIV and SIVcpz isolates. Based on these differences, authors next surveyed gorilla sequences and identified several CD4 haplotypes, specifically in the region encoding the CD4 D1 domain, which directly contacts the viral glycoprotein and thus may impact the interaction. Consistent with this possibility, authors demonstrated that gorilla CD4 haplotypes are, on average, less capable of supporting entry than human CD4, and that some are largely unable to function as SIV entry receptors. Interestingly, individual residues found at key positions in the gorilla CD4 D1 when tested in the context of human CD4 reduce entry of some virions pseudotyped with diverse SIVcpz envelopes, suggesting that individual amino acids can in part explain the observed differences across gorilla CD4 haplotypes. Finally, the authors perform statistical tests to infer that CD4 from great apes with endemic SIV (i.e., chimpanzees and gorillas) but not non-reservoirs (i.e., orangutans, bonobos) or recent spillover hosts (i.e., humans), have been subject to selection as a result of pressure from endemic SIV.

The conclusions of this paper are mostly well supported by data.

Strengths:

(1) The functional assays are appropriate to test the stated hypothesis, and the authors use a broad diversity of envelopes from HIV and SIVcpz strains. Authors also partially characterize one potential mechanism of gorilla CD4 resistance - receptor glycosylation at the derived N15 found in 5/6 gorilla haplotypes.

(2) Ancestral reconstruction provides a particularly interesting aspect of the study, allowing authors to infer the ancestral state of hominid CD4 relative to modern CD4 from gorillas and chimpanzees. This, coupled with evidence supporting SIV-driven selection of gorilla CD4 diversity and the characterization of functional diversity of extant haplotypes provides several interesting findings.

Weaknesses:

(3). The major inference of the work is that SIV infection of gorillas drove the observed diversity in gorilla CD4. This is supported by the majority of SNPs being localized to the CD4 D1, which directly interacts with envelope, and the demonstrated functional consequences of that diversity for viral entry. However, SIVgor (to the best of my knowledge) only infects Western lowland gorillas (Gorilla gorilla gorilla), and one Gorilla gorilla diehli and three *Gorilla beringei* graueri individuals were included in the haplotype and allele frequency analyses. The presence of these haplotypes or the presence of similar allele frequencies in Eastern lowland and mountain gorillas would impact this conclusion. It would be helpful for the authors to clarify this point.

(4) The authors appear to use a somewhat atypical approach to assess intra-population selection to compensate for relatively small numbers of NHP sequences (Fig. 6). However, they do not cite precedence for the robustness of the approach or the practice of grouping sequences from multiple species for the endemic vs other comparison. They also state in the methods that some genes encoded in the locus were removed from the analysis "because they have previously been shown to directly interact with a viral protein." This seems to undercut the analysis, and prevents alternative explanations for the observed diversity in CD4 (e.g., passenger mutations from selection at a neighboring locus).

(5) Data in Figure 5 is graphed as % infected cells instead of virus titer (TDU/mL). It's unclear why this is the case, and prevents a comparison to data in Figure 2 and Figure 4.

(6) The lack of pseudotyping with SIVgor envelope is a surprising omission from this study, that would help to contextualize the findings. Similarly, building gorilla CD4 haplotype SNPs onto the hominin ancestor (as opposed to extant human CD4) may provide additional insights that are meaningful towards understanding the evolutionary trajectory of gorilla CD4.

Comments on revised version:

In the revised manuscript, the authors more appropriately contextualize conclusions that can be made based on their data versus inferences, which are now much more clearly described in the discussion. The authors also included more references to substantiate claims, additional description of methodology, and provided well-reasoned responses to the weaknesses described in my primary review.

Re: #3. As the authors point out, we do not know if eastern gorillas were at one time exposed to SIV. The authors use a variety of phylogenetic and functional approaches to infer that SIVcpz is the selective pressure-shaping gorilla CD4. While I agree this is a highly likely scenario, the allelic diversity of CD4 across gorilla subpopulations raises multiple evolutionary scenarios consistent with the data.

Re: #4. The explanation provided by the authors is reasonable. However, a demonstration that this approach is robust to potential factors that might skew the data (e.g., recombination) is argued but not tested. Part of the concern here is that the study is limited by very small sample sizes, and to the best of my knowledge, grouping sequences from multiple species to make claims about selection is not an established practice. The authors note in their response that they confirmed the existence of CD4 alleles in this study with those identified in 100 gorilla individuals from Russell et al. 2021 (unavailable to the authors at the time of submission) - a re-analysis that includes that data from Russell et al. 2021 would have strengthened the analyses.

---

## [Referee Report · Reviewer #2 (Public Review)]

Lentiviral infection of primate species has been linked to the rapid mutational evolution of numerous primate genes that interact with these viruses, including genes that inhibit lentiviruses as well as genes required for viral infection. In this manuscript, Warren et al. provide further support for the diversification of CD4, the lentiviral entry receptor, to resist lentiviral infection in great ape populations. This work builds on their prior publication (Warren et al. 2019, PMCID: PMC6561292) and that of other groups (e.g., Russell et al. 2021, PMCID: PMC8020793; Bibollet-Ruche et al. 2019, PMCID: PMC6386711) documenting both sequence and functional diversity in CD4, specifically within (1) the CD4 domain that binds to the lentiviral envelope and (2) great ape populations with endemic lentiviruses. Thus, the paper's finding that gorilla populations exhibit diverse CD4 alleles that differ in their susceptibility to lentiviral infection is well demonstrated both here and in a prior publication.

Strengths:

By reconstructing the CD4 sequence from the ancestor of gorillas and chimpanzees, the authors document that modern species have evolved more resistance to (admittedly modern) lentiviruses. They also deconstruct the molecular basis of this resistance by showing that one mutation, which adds a glycosylation site to CD4, is sufficient to confer lentiviral resistance to the susceptible human allele.

Weaknesses:

Warren et al. also pursue two novel lines of evidence to suggest that lentiviruses are the causative driver of great ape CD4 diversification, which seems likely from a logical perspective but is difficult to prove. First, they demonstrate that resistance to lentiviral infection is a derived trait in chimpanzees and gorillas, which have been co-evolving with endemic lentiviruses, but not in humans, which only recently acquired HIV. Nevertheless, these three examples are insufficient to prove that derived resistance is not stochastic or due to drift. The argument would be strengthened by demonstrating that bonobo and orangutan CD4, which also do not have endemic lentiviruses, resemble the ancestral and human susceptibility to great-ape-infecting lentiviruses.

Second, Warren et al. provide a population genetic argument that only endemically infected primates exhibit diversifying selection, again arguing for endemic lentiviruses being the evolutionary driver. The authors compare SNP occurrence in CD4 to neighboring genes, demonstrating that non-synonymous SNP frequency is only elevated in endemically infected species. Moreover, these amino-acid-coding changes are significantly concentrated in the CD4 domain that binds the lentiviral envelope. This is a creative analysis to overcome the problem of very small sample sizes, with very few great ape individuals sequenced. However, the small number of species compared (2-4 in each group) also limits the power of the analysis. Expanding the analysis to Old World Monkey species that do or do not have endemic lentiviruses, as well as great apes, would strengthen this argument.

Overall, this manuscript lends additional support to a well-documented example of a host-virus arms race: that of lentiviruses and the viral entry receptor.

---

## [Author Response]

The following is the authors’ response to the original reviews.

We are thankful for the comments and suggestions from the Editor and Reviewers about our manuscript submitted to the eLife Journal. We have addressed all the comments, and we think these modifications will help bring clarity to our message and be helpful to your readership. Here we include an outline of the corrections performed, as well as a detailed response to each of the reviewer’s comments.

As per the Editor and Reviewers suggestions, outline of corrections:

· The title of the manuscript has been changed to reflect a more conservative conclusion.

· Changes in the main manuscript text were made to enhance clarity, including the use genetic terminology and naming.

· Specific responses to some comments from the reviewers are included in this document. We combined some comments that would be better addressed together.

Accompanied to this letter is an updated version of our manuscript with the track changes feature enabled. Again, we are thankful of the comments and suggestions we received, and we hope this revised version of our manuscript will be accompanied by an updated assessment and public reviews and a final eLife Version of Record.

**Response to the public review and minor recommendations.**

**From Reviewer #1:**
The major inference of the work is that SIV infection of gorillas drove the observed diversity in gorilla CD4. This is supported by the majority of SNPs being localized to the CD4 D1, which directly interacts with the envelope, and the demonstrated functional consequences of that diversity for viral entry. However, SIVgor (to the best of my knowledge) only infects Western lowland gorillas (Gorilla gorilla gorilla), and one Gorilla gorilla diehli and three *Gorilla beringei* graueri individuals were included in the haplotype and allele frequency analyses. The presence of these haplotypes or the presence of similar allele frequencies in Eastern lowland and mountain gorillas would impact this conclusion. It would be helpful for the authors to clarify this point.
**From Reviewer #1 (minor comment):**
Which subspecies of gorilla are the nsSNPs coming from? Gorilla gorilla diehli [n = 1]; *Gorilla beringei* graueri [n = 3] are not extant reservoirs of SIV and to my knowledge are not thought to have been, and so it's important to point out where the diversity is coming from if the authors are asserting that SIVgor drove this population-level diversity in gorilla CD4.

We initially included genomic data from all the gorilla individuals available to maximize sensitivity to identify allelic variants. Although evidence points to eastern gorillas not being currently infected with SIV, our results show that all allelic variants identified have differential susceptibility to the HIV-1 and SIVcpz strains tested. The allelic variants we identified with this genomic data set match the variants identified by Russell et al (doi.org/10.1073/pnas.2025914118), including the ones found in eastern gorillas, and recapitulate that those variants have differential susceptibility to lentiviral entry, similar to the variants of western populations. Whether eastern gorillas have been exposed to lentiviruses in the past remains unknown.

**From Reviewer #1:**
The authors appear to use a somewhat atypical approach to assess intra-population selection to compensate for relatively small numbers of NHP sequences (Fig. 6). However, they do not cite precedence for the robustness of the approach or the practice of grouping sequences from multiple species for the endemic vs other comparison. They also state in the methods that some genes encoded in the locus were removed from the analysis "because they have previously been shown to directly interact with a viral protein." This seems to undercut the analysis and prevents alternative explanations for the observed diversity in CD4 (e.g., passenger mutations from selection at a neighboring locus).

Given the nature of our samples, to detect any influence of natural selection acting on CD4, we chose to compare patterns of molecular evolution of CD4 to its neighboring loci. Comparisons of molecular evolution signatures across genomic regions are the basis of methods to detect positive selection (e.g., Sabeti DOI: 10.1038/nature01140). For our comparison, the neighboring loci represent our neutral standard for the genomic region CD4 resides. Our rationale is that demographic and neutral influences on the number and frequency of polymorphic sites in a region would equally affect all loci in a genomic region. Because these neighboring loci are our neutral benchmark, we excluded before analysis other genes in this genomic region that interact with viruses. The logic is that these loci may be evolving under the influence of positive selection and would decrease the power of our comparison. None of the excluded loci are direct neighbors to CD4. This, and given that the CD4 genomic region in humans is of average recombination rate, dampens the possibility that what we are observing at CD4 is due to selection acting at a neighboring locus. In addition, the classic population genetic method to detect positive selection, the McDonald-Kreitman test (McDonald DOI: 10.1038/351652a0), was originally presented combining polymorphism data across species. We assume that any effect on levels of diversity created by combining variability between species would equally affect all loci included in the study, not just CD4.

**From Reviewer #1:**
Data in Figure 5 is graphed as % infected cells instead of virus titer (TDU/mL). It's unclear why this is the case, and prevents a comparison to data in Figure 2 and Figure 4.
**From Reviewer #1 (minor comment):**
Figure 5: the data presentation is now shown as % infected cells instead of viral titer. This makes it difficult to compare data from Figure 5 to other figures. Can the authors please either justify this change, display data consistently or provide matched data displays as a Supplemental Figure?

For the experiments presented in figures 2 and 4 we used different volumes of infecting pseudoviruses, which allowed us to identify the linear range of infection. Then, based on the number of cells plated per experimental replicate, we calculated a virus titer. In follow-up experiments (Fig. 5), we used fixed volumes of virus that would infect ~10-20% of control (wild-type; wt) CD4-expressing cells. Comparisons were then made between wt and mutated CD4s, and these data are best presented in their raw forms as percent cells infected. Although this change in method prevents direct comparison between the figures, we focused on the differences observed between the experimental conditions per experimental panel.

**From Reviewer #1:**
The lack of pseudotyping with SIVgor envelope is a surprising omission from this study, that would help to contextualize the findings.
**From Reviewer #2 (minor comment):**
The inclusion of HIV-1 but not SIVgor strains in Figures 2D/E is somewhat conspicuous since chimpanzee alleles certainly differ in susceptibility to SIVcpz (and SIVgor) strains per Russell et al. 2021. The authors should either test some SIVgor infections, cite published data on at least extant human/chimpanzee/gorilla CD4 susceptibility to SIVgor, or address why they did not include it.

We agree the data of host susceptibility to SIVgor strains would have been an interesting question to explore. However, we opted to focus on the transmission of SIVcpz strains into gorilla populations for this study. It is worth mentioning that we have cloned SIVgor envelope genes from some strains into our expression system, but we were unable to recover infectious pseudoviruses using an HIV-1DEnv-GFP backbone. This suggests that HIV-1 may be incompatible with incorporating SIVgor Env into virus particles. Recently, Russell et al (DOI: 10.1073/pnas.2025914118) managed to generate SIVgor Env pseudotyped virions using a different backbone (SIVcpzDEnv-GFP) that was unavailable to us at the time of this study.

**From Reviewer #1:**
Similarly, building gorilla CD4 haplotype SNPs onto the hominin ancestor (as opposed to extant human CD4) may provide additional insights that are meaningful toward understanding the evolutionary trajectory of gorilla CD4.

We decided to use the extant human CD4 as a backbone to test the effects on the individual amino acid variants found in the allelic diversity of the gorilla population since the human protein is highly susceptible to all the HIV-1 and SIV strains tested, and the expected phenotype is a loss-of-function. Since the D1 of the human and ancestral sequences for CD4 are almost identical (except for a change that is fixed in gorillas), and they showed similar levels of susceptibility to lentivirus entry, we expect that the phenotypes found would be the same if the gorilla SNPs were built into the ancestral CD4 backbone.

**From Reviewer #2:**
To bolster the argument that lentiviruses are indeed the causative driver of this diversification, which seems likely from a logical perspective but is difficult to prove, Warren et al. pursue two novel lines of evidence. First, the authors reconstruct ancestral CD4 genes that predate lentiviral infection of hominid populations. They then demonstrate that resistance to lentiviral infection is a derived trait in chimpanzees and gorillas, which have been co-evolving with endemic lentiviruses, but not in humans, which only recently acquired HIV. Nevertheless, the derived resistance could be stochastic or due to drift. This argument would be strengthened by demonstrating that bonobo and orangutan CD4, which also do not have endemic lentiviruses, resemble the ancestral and human susceptibility to great-ape-infecting lentiviruses.
**From Reviewer #2 (minor comment):**
The data presented in Figure 2, showing that chimp and gorilla (but not human) CD4 resistance to lentiviral infection is a derived trait, is very intriguing for suggesting that endemic lentiviruses are the causative driver of CD4 evolution. Nevertheless, this could be stochastic or due to genetic drift. Given the later emphasis on several other non-endemically infected species, the authors should at the very least include the sequences for bonobo and orangutan CD4 in the presented alignment (Fig 2B). Ideally, they would also test these orthologs to demonstrate that they are not resistant to lentiviruses infecting great apes (SIVcpz / HIV-1 / SIVgor). If they have also derived resistance, this would suggest a possible other evolutionary driver or genetic drift.

Based on our analysis on polymorphic sites using available data from populations of apes, we strongly believe the accumulation of resistant polymorphisms in CD4 did not arise in a stochastic manner. The frequency and accumulation of these changes strongly correlate with the function of CD4 as a receptor for lentivirus entry. We agree that experimentally testing the CD4 protein from bonobo and orangutan would strengthen our conclusions; however, based on our genomic analyses, we decided to focus on the species that would present a higher level of variability of susceptibility to the lentivirus tested, namely gorillas and chimpanzees.

**From Reviewer #2:**
Warren et al. provide a population genetic argument that only endemically infected primates exhibit diversifying selection, again arguing for endemic lentiviruses being the evolutionary driver. The authors compare SNP occurrence in CD4 to neighboring genes, demonstrating that non-synonymous SNP frequency is only elevated in endemically infected species. Moreover, these amino-acid-coding changes are significantly concentrated in the CD4 domain that binds the lentiviral envelope. This is a creative analysis to overcome the problem of very small sample sizes, with very few great ape individuals sequenced. The additional small number of species compared (2-3 in each group) also limits the power of the analysis; the authors could consider expanding their analysis to Old World Monkey species that do or do not have endemic lentiviruses, as well as great apes.

The scope of this project was to evaluate the differential phenotype of the accumulated polymorphisms found in the ape branch of the primates. Although evaluating the accumulation of polymorphisms in a broader range of primates would generate interesting observations, this would likely require increasing the total number of primate species to include sampling along the speciation tree, many of which lack population level data.

**From Reviewer #1 (minor comment):**
Ancestral reconstruction methods and associated data tables should be included to indicate statistical support for assigned codons. A comment on ambiguity at relevant positions is needed. Similarly, given the polymorphic nature of gorilla and chimpanzee CD4, how confident are the authors in their ancestral reconstructions based on a single representative genome per species? Does this change when you include the broader panel of gorilla sequences? Is the ancestral reconstruction robust to other methods besides PAML?

We used the PAML software package to reconstruct the ancestral hominin and hominid sequence of CD4 because it is a standard and well recognized method for this purpose. For this analysis, we used the set of primate sequences selected for positive selection analyses (see methods), namely the longest isoform sequences for each of the available species that best aligned with human CD4. We feel that the best way to perform to the ancestral state reconstruction was to use only these curated sequences instead of the population level sequences, removing potential biases introduced by having different numbers of variants per species.

**From Reviewer #1 (minor comment):**
Page 10: "It seems that allele 2, which doesn't have this glycan, would be at a fitness disadvantage. In support of this, allele 2 is one of the least frequent alleles in the gorilla population that we surveyed (Figure 3B)." - this inference depends on the gorilla species that encode allele 2 and allele frequencies. There are statistical tests to address this inference.

Population genetic statistics that test for skews in sample allele frequencies are not appropriate here due to the nature of the samples in this study. However, the reviewer is correct that our inference in allele frequency is dependent on the gorilla species that we find this allele in. Allele 2 is found in the *Gorilla beringei* graueri subspecies of gorilla included in this study. We only have data for three individuals (six alleles) from this subspecies compared to 51 individual (102 alleles) from Gorilla gorilla gorilla. As such, genetic subdivision between the gorilla subspecies could also produce the low frequency of allele 2 observed in our sample.

**From Reviewer #1 (minor comment):**
Page 11: "These results imply that the resistance to SIVcpz found in gorilla individuals is not dependent on single amino acids, but rather the cumulative effect of multiple SNPs." Would it be more relevant (or relevant in other ways) to test this statement by putting those mutations into the hominid ancestor? Testing individual residues in the context of human CD4 may be subject to epistasis or several other factors.

We agree that constructing multiple of the resistant SNPs in the susceptible human background would have strengthened our hypothesis, as all these amino acid changes are associated with increased resistance to at least one of the lentiviruses tested. However, the number of CD4 variants to test would increase significantly and we feel that this approach was out of the scope of this manuscript.

**From Reviewer #1 (minor comment):**
Figure 6: If you perform this analysis on chimpanzee CD4 alone do you get the same result? Just gorillas? If you remove eastern/mountain gorillas? The very small numbers of non-human non-SIV-reservoir great apes may preclude a strong conclusion.

We agree that our study is limited by the small number of available sequences from individuals of the studied species. If we remove a whole species or subspecies the statistical power would be greatly reduced. Removing all chimpanzees or gorillas (or a subspecies) would still show that only each of those species accumulate SNPs in the D1 region of CD4, although with less statistical significance.

**From Reviewer #2 (minor comment):**
Related to Figure 2: It would strengthen the argument that resistance is a derived trait if the authors mapped the causative mutations from gorilla CD4 onto the ancestral hominin CD4. However, this experiment is not particularly critical, merely a suggestion.

We appreciate this suggestion. We decided to use the human CD4 backbone as it is widely susceptible to lentiviral entry. The hominid and hominin ancestral sequences are almost identical to the human sequence in domain 1, except for a fixed mutation shared with the gorilla CD4. We expect that the SNPs observed in the gorilla population would also reduce susceptibility to lentivirus entry in the ancestral CD4 reconstructions.

**From Reviewer #2 (minor comment):**
Related to Figure 3B: It is difficult to make much of the allele frequency for 8 alleles in 32 individuals. Can the authors collate this with allele frequency for the referenced 100 individuals from Russell et al. 2021, to give a better sense of population frequency? This may allow the authors to better correlate allele frequency with SIVcpz resistance patterns in Figure 4, strengthening their argument that more resistant alleles should be over-represented in the population.

At the time of our analysis the data from Russell (DOI: 10.1073/pnas.2025914118) was not available to collate or compare. When that data became available, we immediately compared the existence of the alleles found and confirmed that the ones we found were also detected in the samples used in that study.

**From Reviewer #2 (minor comment):**
Related to Figure 6: As written, several methodological details should be clarified. How were human genomes selected to limit the sample size to 50?

We selected a total of 50 human individuals in order to size-match the sample size of the largest group in Fig 6B (chimpanzee, n=50). We randomly selected 10 individuals for each of the 5 superpopulations [Africans (AFR), Admixed Americans (AMR), East Asians (EAS), Europeans (EUR) and South Asians (SAS)] defined by the 1000 Genome Project.

**From Reviewer #2 (minor comment):**
Related to Figure 6: What comparison is being reported for the Mann-Whitney U test (CD4 vs. which gene)? Are the means shown in A an average of 2 (endemic) or 3 (non-endemic) species - if so, the authors should show the individual data points to give a clearer depiction of the data spread. In addition, it is not clear that a statistical test with sample sizes of 2 is meaningful, since Mann Whitney typically assumes n > 5. To strengthen this statistical argument, it may be necessary to include additional species that have (a) multiple genomes (or at least this locus) sequenced, and (b) have or lack lentiviral sequences. This may necessitate expanding the analysis to include Old World Monkeys (e.g. Rhesus Macaque Genome Project).

In the Figure 6 we use the Mann-Whitney U test to compare variation between CD4 and the neighboring loci. The average and SEM are for two endemic and four non-endemic species (two orangutan datasets are from two distinct species vs the gorilla subspecies). It is true our sample size is small for any statistical testing. For the Mann-Whitney U-test it is generally preferred to have n > 5 in each group. So, we do run into problems with the endemically infected comparisons as we only have two data points (chimpanzee and gorilla) for the CD4 group. For the uninfected species, CD4 has four data points.

**From Reviewer #1 (minor comment):**
Page 6. "This suggests that the ancestral versions of CD4 in apes were susceptible to primate lentivirus entry" - The data show that tested virus pseudotyped with SIV/HIV envs can engage ancestral CD4 in the context of a canine cell line expressing human CCR5, but not necessarily that this interaction was sufficient for the process of entry per se, especially in the context of a gorilla (or hominid) cell. Some additional context would be useful for a broad readership.
**From Reviewer #1 (minor comment):**
Page 6: "but that selective pressures exerted by SIVs in the chimpanzee and gorilla lineages have led to the retention of mutations that confer resistance to primate lentivirus infection. This has not happened in humans where selective pressure by HIV-1 is too new" - this cannot be concluded from the data in Figure 1. It would be more appropriate as a Discussion point.
**From Reviewer #1 (minor comment):**
Page 14: "Natural tolerance is often required before a virus can establish itself long term in a host reservoir, and thus understanding it is key to understanding virus reservoirs in nature" - please provide a reference. This is one among several theories of long-term host-virus evolution dynamics/outcomes, and further discussion may benefit the broad readership of eLife.
**From Reviewer #1 (minor comment):**
Page 15: "There is a surprising outcome of virus-driven host evolution in that the divergence and diversity of these host genes ultimately comes at a detriment to the very viruses that drove this evolution." - it is not clear to this reviewer why this is surprising.
**From Reviewer #2 (minor comment):**
Related to Figure 5A: The authors suggest that the gorilla glycosylation site provides resistance to SIVcpz, based on TAN1.910, but in fact the glycosylated allele is no more resistant than the un-glycosylated allele to most SIVcpz strains (in Figure 4). The authors should acknowledge this more clearly in the text.
**From Reviewer #2 (minor comment):**
The title of this article (that infection "has driven selection") is somewhat overstated - though it seems very likely that lentiviruses are driving CD4 diversification, this is difficult to prove. The arguments presented here rely on very few data points: modern chimp and gorilla compared to ancestral CD4, and a population genetic analysis relying on 2 or 3 species with 10-50 individuals each. The authors should either bolster these arguments (see the above suggestions) and/or soften the claim in the title.

Modifications to the main text of the manuscript have been made to enhance clarity on the subjects stated above.